# Taming the waves: sine as activation function in deep neural networks

**Giambattista Parascandolo, Heikki Huttunen & Tuomas Virtanen**
Department of Signal Processing
Tampere University of Technology
Tampere, Finland
{giambattista.parascandolo,heikki.huttunen,tuomas.virtanen}@tut.fi

## Abstract

Most deep neural networks use non-periodic and monotonic—or at least quasiconvex— activation functions. While sinusoidal activation functions have been successfully used for specific applications, they remain largely ignored and regarded as difficult to train. In this paper we formally characterize why these networks can indeed often be difficult to train even in very simple scenarios, and describe how the presence of infinitely many and shallow local minima emerges from the architecture. We also provide an explanation to the good performance achieved on a typical classification task, by showing that for several network architectures the presence of the periodic cycles is largely ignored when the learning is successful. Finally, we show that there are non-trivial tasks—such as learning algorithms—where networks using sinusoidal activations can learn faster than more established monotonic functions.

## 1 Introduction

Most activation functions typically used nowadays in deep neural networks—such as sigmoid, tanh, ReLU, Leaky ReLU, ELU, parametric ReLU, maxout—are non-periodic. Moreover, these functions are all quasiconvex, and more specifically either monotonic (sigmoid, tanh, ReLU, Leaky ReLU, ELU) or piece-wise monotonic with two monotonic segments (parametric ReLU, maxout).

Monotonicity makes sense from an intuitive point of view. At any layer of a network, neurons learn to respond to certain patterns, i.e. those that correlate with their weights; in case of monotonic functions, to a stronger positive correlation corresponds a stronger (or equal) activation, and viceversa, to a weaker positive correlation corresponds a weaker (or equal) activation. Neurons using piece-wise monotonic functions with two monotonic segments can be viewed as two separate neurons, each equipped with one of the two monotonic segments, and therefore independently looking for either the positive or the negative correlation between the weights and the input.

Excluding the trivial case of constant functions, periodic functions are non-quasiconvex, and therefore non-monotonic. This means that for a periodic activation function, as the correlation with the input increases the activation will oscillate between stronger and weaker activations. This apparently undesirable behavior might suggest that periodic functions might be just as undesirable as activation functions in a typical learning task.

But is this really the case? As shown in Section 2, there are several examples from the literature where sinusoidal functions were successfully used in neural networks. Moreover, as noted already in Gaynier & Downs (1995), networks using simple monotonic activation functions—such as sigmoids, tanh, ReLU—tend to have smaller VC dimension than those using non-monotonic functions. More specifically, even a network with a single hidden neuron using sinusoidal activation has infinite VC dimension[1].

Neural networks using sinusoidal activation functions have been regarded as difficult to train (Lapedes & Farber (1987)) and have been largely ignored in the last years. There are a few questions

---

[1] I.e., it can correctly classify any set of points.

that naturally arise and make an analysis of deep neural networks using periodic activation functions interesting:

- What makes them in theory difficult to train?
- Why do they still often manage to learn in practice?
- How does the learned representation differ from the one of similar quasi-convex functions?
- Are there tasks where periodic activation functions are more apt than quasiconvex ones?

In this paper we shed some light on these questions. In Section 2 we review relevant works on the topic of periodic activation functions. Starting from a simple example, in Section 3 we show what makes learning with sinusoidal activations a challenging task. In Section 4 we run a series of corroborative experiments, and show that there are tasks where sinusoidal activation functions outperforms more established quasi-convex functions. We finally present our conclusions in Section 5.

## 2 RELATED WORK

Periodic activation functions, and more specifically sinusoids, have received a tiny fraction of the attention that the research community reserved to the more popular monotonic functions. One of the first notions of a neural network with one hidden layer using sine as activation comes from (Lapedes & Farber, 1987, pp. 25-26). The authors define it as a *generalized Fourier decomposition*, and while recognizing the potential in their approximation capacity, they report that in their experiments these networks often exhibited numerical problems or converged to local minima.

In Sopena et al. (1999) the authors show on several small datasets that a multi layer perceptron with one hidden layer using sinusoids improves accuracy and shortens training times compared to its sigmoidal counterpart. For similar networks, improvements are shown in Wong et al. (2002) for a small handwritten digit recognition task and in McCaughan (1997) for the validity of logical arguments.

Some works have concentrated on mixing periodic and non periodic activations. In Fletcher & Hinde (1994) the authors propose to learn a coefficient that weighs each activation between sine and sigmoid. More recently, in Gashler & Ashmore (2016) the authors used sinusoids, linear and ReLU activations in the first layer of a deep network for time-series prediction.

Some theoretical results were presented in Rosen-Zvi et al. (1998), where the authors analyze the learning process for networks with zero or one hidden layers, and sinusoidal activations in all layers. In Nakagawa (1995) the author shows that a chaotic neuron model using a periodic activation function has larger memory capacity than one with a monotonous function.

Concerning recurrent neural networks (RNNs), in Sopena & Alquezar (1994) and Alquézar Mancho et al. (1997) the activation function for the last fully connected layer of a simple RNN was sine instead of sigmoid, which led to higher accuracy on a next-character prediction task. Choueiki et al. (1997) and Koplon & Sontag (1997) used sinusoidal activations in a RNN for short-term load forecasting and fitting sequential input/output data respectively. Liu et al. (2016) studied the stability of RNNs using non-monotonic activation functions, trying also sinusoids along others. No work so far—to the best of the authors' knowledge—has investigated the use of periodic activation functions in convolutional neural networks (CNNs).

A separate line of research has focused on networks that closely mimic Fourier series approximations, so called *Fourier series neural networks* (Rafajłowicz & Pawlak (1997); Halawa (2008)). Here the hidden layer is composed of two parts: each input node is connected to an individual set of hidden nodes using sines and cosines as activations. The input-to-hidden connections have independent and fixed weights (with integer frequencies $1...K$) for each input dimension. Then, the product is computed for each possible combinations of sines and cosines across dimensions. After that, only the hidden-to-output connections—which correspond to the Fourier coefficients—are learned. Despite the good theoretical properties, the number of hidden units grows exponentially with the dimensionality of the input (Halawa (2008)), rendering these networks impractical in most situations.

## 3 ANALYSIS OF SINUSOIDAL ACTIVATION FUNCTIONS

Let us start with a definition of the framework studied. In this section we analyze a deep neural network (DNN) with one hidden layer and linear activation at the output. The network receives as input a vector $\mathbf{x}$—that has an associated target $\mathbf{y}$—and computes an hidden activation $\mathbf{h}$ and a prediction $\hat{\mathbf{y}}$ as

$$\mathbf{h} = \mathcal{F}(\mathbf{Wx} + \mathbf{b_W}) \tag{1}$$
$$\hat{\mathbf{y}} = \mathbf{Ah} + \mathbf{b_A} \tag{2}$$

where $\mathbf{W}$ and $\mathbf{A}$ are weight matrices, $\mathbf{b_W}$ and $\mathbf{b_A}$ are bias vectors, and $\mathcal{F}$ is an activation function. As noted already in previous works, there is a clear interpretation of the variables in the network when $\mathcal{F} = \sin$, in terms of a Fourier representation. The weights $\mathbf{W}$ and the biases $\mathbf{b_W}$ are respectively the frequencies and phases of the sinusoids, while $\mathbf{A}$ are the amplitudes associated, and $\mathbf{b_A}$ the DC term. As shown in Cybenko (1989); Jones (1992) such a network can approximate all continuous functions on $C(I_n)$, i.e. on the $n$-dimensional hypercube.

### 3.1 LEARNING WITH SINES AND LOCAL MINIMA

We can encounter issues with local minima even when learning the network parameters to solve a very simple optimization problem. Let us assume we are trying to learn the target function $g(x) = \sin(\nu x)$ for $-m < x < m$ and some frequency $\nu \in \mathbb{R}$. $x$ is the input to the network, and for this analysis we treat the case of continuous and uniformly distributed data, but we argue later in the section that similar conclusions can be expected with a limited amount of randomly distributed samples. By training a network with a single hidden neuron, fixed hidden-to-output connection $\mathbf{A} = a = [1]$ and no biases, i.e. no phase nor DC term to learn, our problem is reduced to learning the frequency $\nu$ as the weight $\mathbf{W} = [w]$.

Formally, we are minimizing the squared loss $(\sin(\nu x) - \sin(wx))^2$. For a fixed choice of $\nu$ and $m$, the loss landscape $\mathrm{L}(\nu, w, m)$ wrt to $w$ has the form

$$
\begin{aligned}
\mathrm{L}(\nu, w, m) &= \int_{-m}^{m} (\sin(\nu x) - \sin(wx))^2 dx \\
&= -\frac{2 \sin(m(w - \nu))}{w - \nu} + \frac{2 \sin(m(w + \nu))}{w + \nu} - \frac{\sin(2mw)}{2w} + c(\nu, m)
\end{aligned}
\tag{3}
$$

where $c(\nu, m)$ is a constant term. As illustrated in Fig. 1, for a fixed choice of $\nu$ and $m$, the three main terms in $\mathrm{L}(\nu, w, m)$ are three cardinal sines (or sincs): the first is negative and centered at $w = \nu$, which is the only global minimum and where the loss is 0; the second term is positive and centered at $w = -\nu$, and is the only global maximum; the third sinc is negative and centered in $w = 0$. The latter creates a local minimum for small values of $w$ and large values of $m$ and $\nu$, where the function expressed by the network is a constant $\sin(0) = 0$.

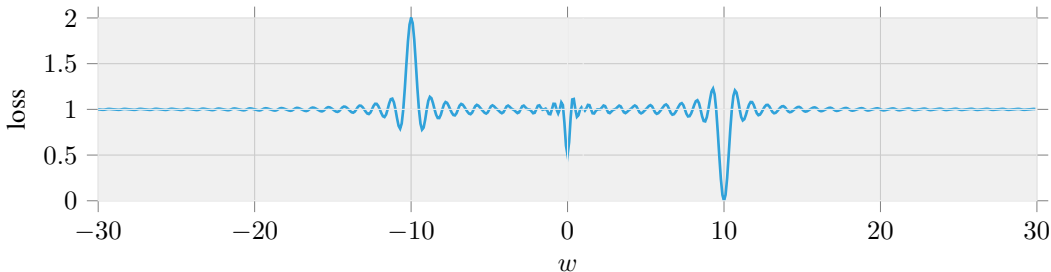

Figure 1: The loss surface when only the frequency $\nu = 10$ of a sine needs to be learned. One of the three sincs is centered in 0, the other two $w = \pm 10$.

We can already spot two culprits responsible for the difficulty of this learning problem:

(i) the deep local minimum centered in $w = 0$, produced by the sinc centeed in 0, which "traps" small weights around zero

(ii) the infinitely many ripples created by all three sincs, each of which is a shallow local minimum.

Also note that away from the main lobes the overall shape of the loss is almost flat, and therefore if the optimization starts far from the global optimum the gradients will tend to be small.

Let us now make the result more general, by including again the amplitudes and bias terms, and trying to learn a more complex function. After adding a bias/phase term to the neuron and to the target function $g(x)$ ($b$ and $\phi$ respectively) and a hidden-to-output weight/amplitude term, ($a$ and $\gamma$ respectively), we are trying to minimize $(\gamma \sin(\nu x + \phi) - a \sin(wx + b))^2$. From the solution of the integral, the equation describing the second summand in Eq. 3 gains a term $a\gamma \cos(b + \phi)$, while the third summand gains a term $a^2 \cos(2b)$. Therefore all the sincs are still present (as shown in Fig. 2), and so are the aforementioned side effects.

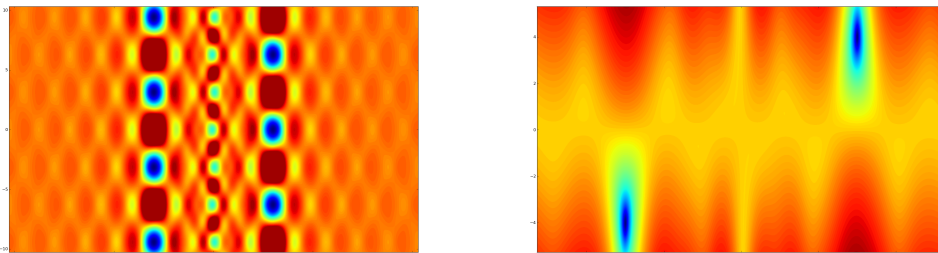

(a) $y$-axis is the phase $b$, $x$-axis the frequency $w$ (b) $y$-axis is the amplitude $a$, $x$-axis the frequency $w$

Figure 2: The loss surface as a function of the network parameters when trying to learn $g(x) = 1 \sin(\nu x + 0)$. Cold colors are smaller values. The local minima in the ripples generated by the sincs are clearly visible.

Moreover, the local minimum centered in zero comes from the integral

$$\int \sin^2(wx)dx = \frac{x}{2} - \frac{\sin(2wx)}{4w} + c, \tag{4}$$

which appears after expanding the square of the sum and applying linearity to the integral in $L(\nu, w, m)$. Note that this term is not related to the function to be learned $g(x)$, nor to the fact that there is a single hidden neuron, and therefore will always appear in any network with a single layer of sinusoids trained using mean squared error.

Finally, since any function in the class that we are considering can be approximated to desired precision using a finite sum of sinusoids $g(x) \approx \sum_{i=0}^{M} \gamma_i \sin(\nu_i x + \phi_i)$, we can turn our analysis to any target function $g(x)$. The resulting function to be minimized $\left[\sum_{i=0}^{M} \gamma_i \sin(\nu_i x + \phi_i) - \sum_{i=0}^{N} a_i \sin(w_i x + b_i)\right]^2$ is again the square of the sum of multiple sinusoids. After squaring and applying linearity, every term will either be $\sin^2(\cdot)$ or $\sin(\cdot)\sin(\cdot)$ (with some amplitude terms). The former produces a sinc centered in zero, while the latter an odd pair of sincs.

Despite all this, the problems we just described are typically not an issue for many tasks. Going back to the example with a single sinusoid to learn, we can notice that the central local minimum disappears when the frequency $\nu$ is small enough that the main lobe of the rightmost sinc incorporates the main lobe of the central sinc (see Fig. 3). This happens when the data has a frequency representation with a large amount of low frequencies, which we assume to be often the case for many realistic datasets. The size of the support $m$ also has an effect on the width and depth of the sincs. In a practical case at training time the integral is replaced by a sum—since only a limited amount of training samples is available—, the sampling is typically not uniform, and there might be noise in the data. Moreover, in the analysis we assumed that the loss surface (and therefore the

gradient) is calculated on the full training set, while in practice only mini batches of training samples are typically used. All these factors can contribute to smooth the loss surface L, potentially making the task easier to solve (see Fig. 3).

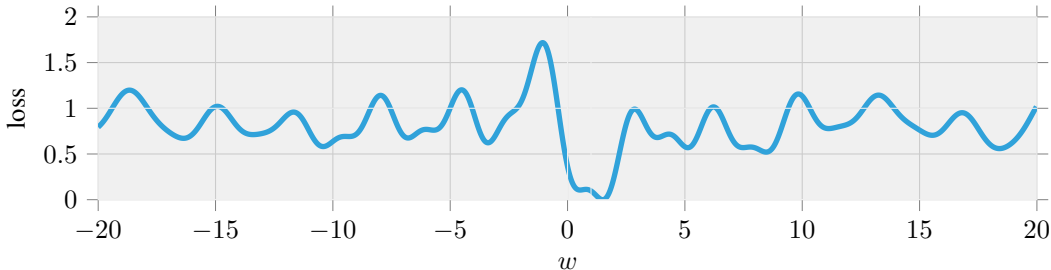

Figure 3: The loss surface when only the frequency of the target sinusoid needs to be learned, only a set of non-uniformly distributed samples is available at training time, and for a low frequency $\nu$ of the target function. Note that the central local minimum has disappeared.

On these premises, we can expect that learning will be difficult when $g(x)$ has large high frequency components (disjoint sincs). If network weights are initialized with small enough values, the weights might remain trapped inside the local minima of the central sincs. For large initialization the network might still be unable to find the global minimum due to the absence of overall curvature and the presence of shallow local minima. The optimization will be hard also if $g(x)$ has low frequency components and the weights are initialized with large values. We speculate that a large initialization of the weights, typical in the past, was the main reason why these networks were regarded as difficult to train even with a single hidden layers.

Extending the analysis to deeper networks using sinusoids is not as simple. Already for two hidden layers the resulting function is of the form $\sin(\sin(\cdot))$, whose integral is not known analytically in closed form.

## 3.2 INITIALIZATION AND USE OF PERIODICITY

As a consequence of the results presented in Section 3.1, the correct initialization range of the weights using sine might be very different from the one used for other activation functions. If the weights are very small, the sinusoid acts in its central linear region (Fig. 4).

While for inherently periodic tasks it is reasonable to assume that the network might indeed perform better, several tasks analyzed in Section 2 are not clearly periodic. None of the aforementioned works has analyzed the possibility that the network used mostly the monotonic segment of the sinusoid around zero, which is very similar to the $\tanh$ (Fig. 4). Especially in the typical training scenario—where the input data $\mathbf{x}$ is normalized to have zero mean and unit variance, and the network initialization is done using small weights $\mathbf{W}$ and zero biases—most pre-activations $\mathbf{z} = \mathbf{W}\mathbf{x} + \mathbf{b}$ are likely to be such that $|z| < \pi/2$.

In Section 4 we run a series of experiments to investigate if and how much a network trained using sine as activation actually relies on the periodic part.

## 4 EXPERIMENTS

In this section we train several networks using $\sin$ as activation function on the MNIST and Reuters dataset. We then investigate how much of the periodicity is actually used by replacing the activation function in the trained network with the truncated $\sin$, (abbreviated as $tr.\sin$), defined as

$$tr.\sin = \begin{cases} 0, & \text{if } -\pi/2 < x \\ \sin(x), & \text{if } -\pi/2 \le x \le \pi/2 \\ 1, & \text{if } x > \pi/2 \end{cases} \qquad (5)$$

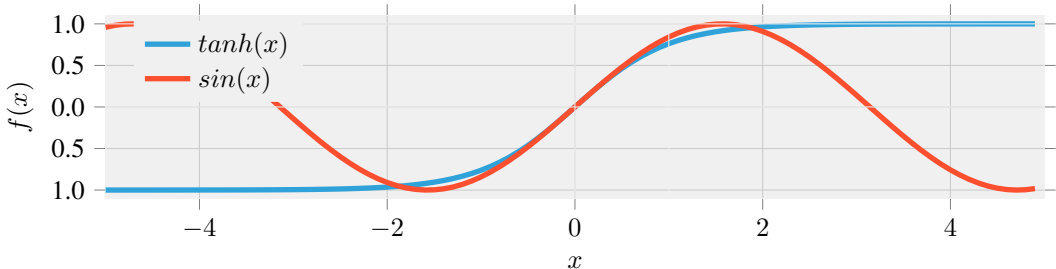

Figure 4: $\sin(x)$ and $\tanh(x)$ are very similar for $-\pi/2 < x < \pi/2$. The network might end up using only this part of the sine, therefore treating it as a monotonic function and ignoring its periodicity.

We also train the same networks using the monotonic function $\tanh$ for comparison. We then run experiments on a couple of algorithmic task where the nature of the problem makes the periodicity of the sinusoid potentially beneficial.

## 4.1 MNIST

We experiment with the MNIST dataset, which consists of 8-bits gray-scale images, each sized 28 x 28 pixel, of hand-written digits from 0 to 9. The dataset has 60,000 samples for training and 10,000 samples for testing. It is simple to obtain relatively high accuracy on this dataset, given that even a linear classifier can achieve around 90% accuracy. Since the data is almost linearly separable, it is reasonable to expect that using sine as activation function will not make much use of the periodic part of the function. We test a DNN, a CNN and an RNN on this problem, using sine as activation function, and compare the results to the same network trained using tanh.

On all experiment on MNIST we scale the images linearly between 0 and 1. All networks have an output layer with 10 nodes, use softmax and are trained with cross-entropy as loss. The batch size is 128 and the optimizer used is Adam (Kingma & Ba (2015)) with the hyper-parameters proposed in the original paper.

**DNN**  We use a DNN with 1 to 2 hidden layers, each with 256 hidden neurons. We initialize the weights in all layers using a normal distribution with standard deviation $\sigma$ in the set 1, 0.1, 0.01. The input images are flattened to vectors of size $28 \times 28 = 784$, which makes the task referred to as *permutation invariant MNIST*. The networks are trained for 20 epochs.

**RNN**  The input images are presented as a sequence of 28 rows, each containing 28 values, starting from the top to the bottom. We use a RNN with 1 hidden layer with 128 hidden neurons. We experiment separately with vanilla RNNs and LSTMs. When the latter are used with sine, the function is used in place of the inner $\tanh$ activation. We initialize the weights in all recurrent layers using a normal distribution with standard deviation of 0.1.

The DNN results are reported in Table 1. As expected, replacing the activation from $\tanh$ to truncated $\sin$ does not affect much the results. For this reason we will not report this value on the following tables. For the same reason, switching $\sin$ to either $\tanh$ or to truncated $\sin$ has almost the same effect, so we will only report the latter from here onwards. When using small values of $\sigma = \{0.1, 0.01\}$ for the initialization, all networks equipped with sines obtained very similar results to the networks trained with tanh. Even though for these networks between 27% to 47% of the activations fall outside of the range $[-\pi/2, \pi/2]$, replacing $\sin$ with $tr. \sin$ does not reduce the accuracy by more than 2.5%. We can therefore conclude that the network is ignoring for the most part the periodicity of the function. On the contrary, the $\tanh$ is more significantly relying on the saturated part, and as $\sigma$ increases so does the drop in the accuracy when switching the activation function to sine (reaching random guessing accuracy for $\sigma = 1$).

As expected from the results presented in Section 3, the networks with sine had difficulty to converge for large initialization $\sigma = 1$[2]. Also notice that adding weight decay allowed the same network with 1 hidden layer to converge, reaching a solution that scarcely uses the periodic part of the function. Finally, results show that even for deeper networks with eight hidden layers, sinusoid can learn the task quite effortlessly, and still does so scarcely relying on the segment of the function outside $[-\pi/2, \pi/2]$.

A somewhat similar but less evident behavior emerged from the RNNs, as shown in Table 2. Especially for the LSTMs, the network using $\tanh$ relied on larger pre-activations much more than the network using $\sin$.

Table 1: MNIST results for DNNs. For each row, we train a network using either $\tanh$ or $\sin$ and report the results on the test data. We then replace the activation in the trained models with the one followed by $\rightarrow$, and directly recompute the accuracy on the test set without retraining the networks. The last column reports the percentage of hidden activations for the $\sin$ networks that exceeds the central monotonic segment of the sinusoid.

| Network | $\tanh$ | $\tanh \rightarrow tr.\sin$ | $\tanh \rightarrow \sin$ | $\sin$ | $\sin \rightarrow \tanh$ | $\sin \rightarrow tr.\sin$ | $\%|z|>\pi/2$ |
|---|---|---|---|---|---|---|---|
| DNN 1-L init 0.01 | 98.0 | 98.1 | 98.0 | 98.0 | 95.2 | 95.6 | 38% |
| DNN 2-L init 0.01 | 98.2 | 98.2 | 81.4 | 98.2 | 95.1 | 95.6 | 27%, 48% |
| DNN 1-L init 0.1 | 98.1 | 98.1 | 78.1 | 98.1 | 96.1 | 96.3 | 47% |
| DNN 2-L init 0.1 | 98.2 | 98.2 | 81.3 | 98.1 | 96.1 | 96.5 | 29%, 47% |
| DNN 1-L init 1 | 95.6 | 95.5 | 10.0 | 16.9 | 13.6 | 13.8 | 86% |
| DNN 2-L init 1 | 92.8 | 92.5 | 10.0 | 10.0 | 10.0 | 10.0 | - |
| DNN 1-L init 1, $10^{-4}$L2 | 96.8 | 92.5 | 10.0 | 97.7 | 96.0 | 96.1 | 14% |
| DNN 8-L init 0.1 | 97.8 | 97.8 | 59.5 | 97.0 | 92.7 | 93.7 | all $\approx$40% |

Table 2: MNIST results for RNN and LSTM.

| Network | $\tanh$ | $\tanh \rightarrow \sin$ | $\sin$ | $\sin \rightarrow tr.\sin$ |
|---|---|---|---|---|
| RNN init 0.1 | 96.3 | 81.3 | 97.4 | 94.1 |
| LSTM init 0.1 | 97.3 | 77.6 | 97.2 | 93.7 |

Similar experiments with the Reuters dataset (ref) showed the same behavior, as seen in table 3. Each sequence of words corresponding to a data sample from the dataset is first converted to a vector of size 1000, where the i*th* entry represents the amount of times that the i*th* most frequent word in the dataset appears in the sentence. The DNN has 128 hidden neurons, networks are trained for 20 epochs and test results are computed on a held-out 20% of the data.

Table 3: Reuters results for DNNs. On the training data all the original architectures — i.e. without changing the activation function after training — reach an accuracy $> 90\%$

| Network | $\tanh$ | $\tanh \rightarrow \sin$ | $\sin$ | $\sin \rightarrow tr.\sin$ |
|---|---|---|---|---|
| DNN 2-L init 0.01 | 75.9 | 71.6 | 76.1 | 76.3 |
| DNN 2-L init 0.1 | 77.0 | 76.0 | 77.3 | 77.9 |
| DNN 2-L init 1 | 61.6 | 3.2 | 16.4 | 8.5 |

## 4.2 LEARNING ALGORITHMIC TASKS

We test the networks using sine as activation on a couple of algorithmic tasks, such as sum or difference of D-digits numbers in base 10. In both tasks the data is presented as a sequence of one-hot encoded vectors, where the size of the vector at each timestep is 12: the first 10 entries correspond to the numbers from 0 to 9, the last two entries correspond to the operator symbol—'+'

---

[2]The network with $\sin$, 1-L and $\sigma = 1$ reaches an accuracy of 40% on the training data after 20 epochs and 83% after 1000 epochs. With two hidden layers it has random guessing accuracy on the training data after 20 epochs, and after 100 epochs 100% accuracy on training data and random guessing accuracy on test data.

or '$-$' in case of sum or difference respectively—and the *'blank'* symbol used for padding. The length of an input sequence is $D + 1 + D$, while the output sequence has length $D + 1$. If a string is shorter than the total length, the remaining entries are padded with the *'blank'* symbol.

For the task *sum* (*difference*) the network is expected to produce the result of the sum (difference) of two positive integers fed as input. We run experiments with the number of digits $D = 8$. The order of the digits of each number is inverted, which was shown to improve the performance in several tasks using encoder-decoder (ENC-DEC) architectures.

We use an encoder-decoder architecture based on vanilla RNN or LSTM. The networks have 128 hidden units in every layer, one recurrent layer for encoding and one recurrent layer for decoding. The decoder has also a fully connected output layer with softmax at each step. The encoder "reads" the input sequence one symbol at a time and updates its hidden state. At the end of the input sequence, the hidden state from the encoder is fed at each step for $D + 1$ times as input to the decoder. The decoder produces the output, one digit at a time.

The networks are trained for 5000 iterations[3] using Adam as optimizer, cross-entropy as loss and a batch size of 128. The feed-forward and recurrent weights are initialized using a normal distribution with the widely used schemes proposed in Glorot & Bengio (2010) and Saxe et al. (2013) respectively, we clip gradients at 1 and decay the learning rate by $10^{-5}$ at every iteration. Samples are generated at each iteration and we do not use a separate validation or test set, since the number of possible samples is so large that overfitting is not an issue. The accuracy for a given prediction is 1 only if every digit in the sequence is correctly predicted. The results reported in Fig. 5 are computed at every iteration on the newly generated samples *before* they are used for training.

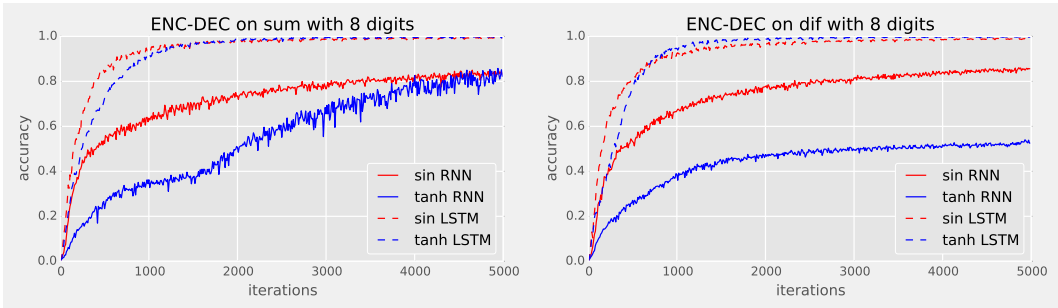

Figure 5: Accuracy curves of the ENC-DEC LSTM and RNN using sine or tanh. The number of digits for each sequence is sampled uniformly in $\{1, ..., D\}$. For uniform sampling of the addends in $\{0, ..., 10^D - 1\}$—which prevents small addends to appear very often—the experiments are in the appendix.

The networks using sine learn the tasks faster and with higher accuracy than those using tanh. While in vanilla RNNs the difference is quite evident, the improvement is less striking for the LSTM. In all the models switching the activation from sine to truncated sine, or from tanh to sine brings the accuracy almost to 0, indicating that the network is effectively using the periodic part of the function.

## 5 CONCLUSIONS

Neural networks with a single hidden layer using sinusoidal activation functions have been largely ignored and regarded as difficult to train. In this paper we analyzed these networks, characterizing the loss surface, and showing in what conditions they are especially difficult to train. By looking into the hidden activations of networks succesfully trained on a simple classification task, we showed that when learning is successful the networks often scarcely rely on the periodicity of the sinusoids.

Finally, we showed on a pair of simple algorithmic tasks where the periodicity is intuitively beneficial, that neural networks using sinusoidal activation functions can potentially learn faster and better than those using established monotonic functions on certain tasks. This encourages future

---

[3]Here we refer to one iteration as the processing of 128 minibatches.

work to investigate the use of periodic functions, the effect at different layers, and the potential of incorporating these functions in other models using quasi-convex functions.

ACKNOWLEDGMENTS

The authors wish to acknowledge CSC IT Center for Science, Finland, for computational resources.

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

# A APPENDIX

We report here similar experiments to Section 4.2 but using uniform sampling of the addends, instead of uniform sampling of the number of digits of each addend.

As shown on the left plots in Fig. 6, for $D = 8$ the networks using $\sin$ learn faster and reach higher accuracy than the network using $\tanh$. For the case of $D = 16$ and 3 recurrent layers in the encoder, sine reaches almost 80% accuracy, while tanh never takes off in the 5000 epochs of training. A similar behavior emerges on the task *dif*, as shown in Fig. 7, although with overall lower accuracy and with none of the networks successfully learning the task with $D = 16$. Surprisingly, the LSTMs almost completely fail to learn the tasks under these training setting.

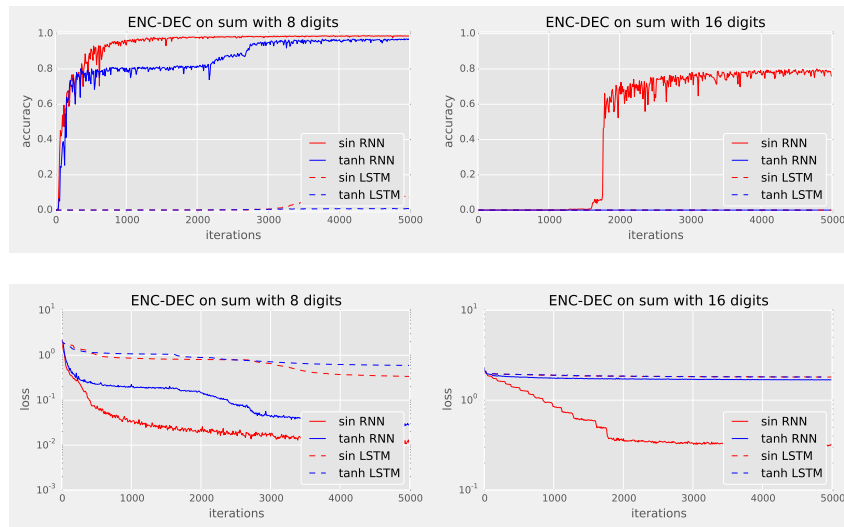

Figure 6: Accuracy and loss curves of the ENC-DEC RNN using sine or tanh, for the task *sum* with 8 or 16 digits per addend. The digits are sampled uniformly.

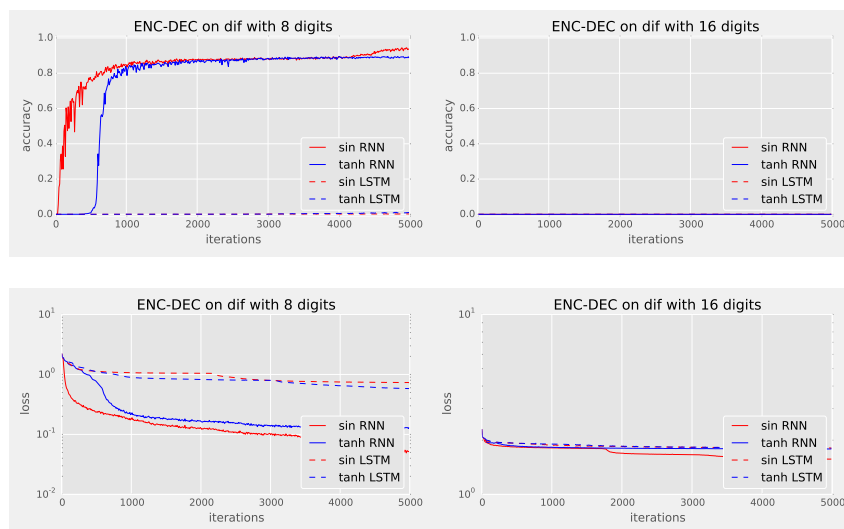

Figure 7: Accuracy and loss curves of the ENC-DEC RNN using sine or tanh, for the task *dif* with 8 or 16 digits per addend. The digits are sampled uniformly.

**More hidden neurons**   When the number of hidden units is doubled from 128 to 256, the standard LSTM using $\tanh$ as activation learns faster and reaches higher accuracy than the network trained with sine, while the vanilla RNN using $\sin$ still outperforms both the vanilla RNN and the LSTM using $\tanh$. These results are reported in Fig. 8 for the case of $D = 8$ only, since for $D = 16$ all networks are stuck at zero accuracy. Further investigation would be required to explain how doubling the amount of neurons in the $\tanh$ LSTM changed the learned representation, providing such a boost in performance.

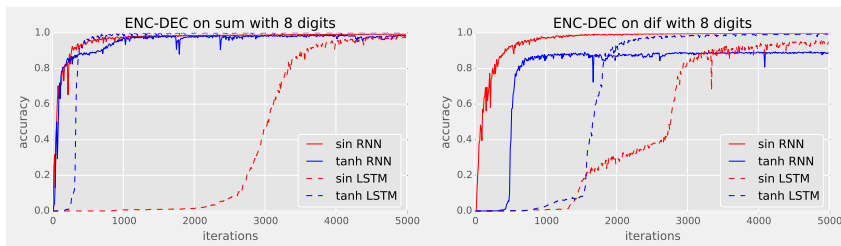

Figure 8: Accuracy curves of the ENC-DEC RNN using sine or tanh, for the tasks *sum* and *dif* with 8 digits per addend. The digits are sampled uniformly.

**Curriculum learning experiments**   For the $D = 16$ case, we also experiment with a curriculum learning approach: while keeping the length of the input and output sequences fixed to 16+1+16 and 17 respectively, we start training by limiting the maximum number of digits to $D = 8$ and increase $D$ by 2 every 1000 iterations, so that by the 4000th iteration $D = 16$. As shown in Fig. 9, by using this approach the network using sine as activation function reaches an accuracy close to 1 by the end of the training. As shown by the steep drops in performance when $D$ is increased, the network has only learned to correctly perform the operation within the number of digits it was trained upon, but it can adapt very quickly to the longer addends. The network using tanh takes more time to learn the case with $D = 8$ and after that does not adapt to larger number of digits.

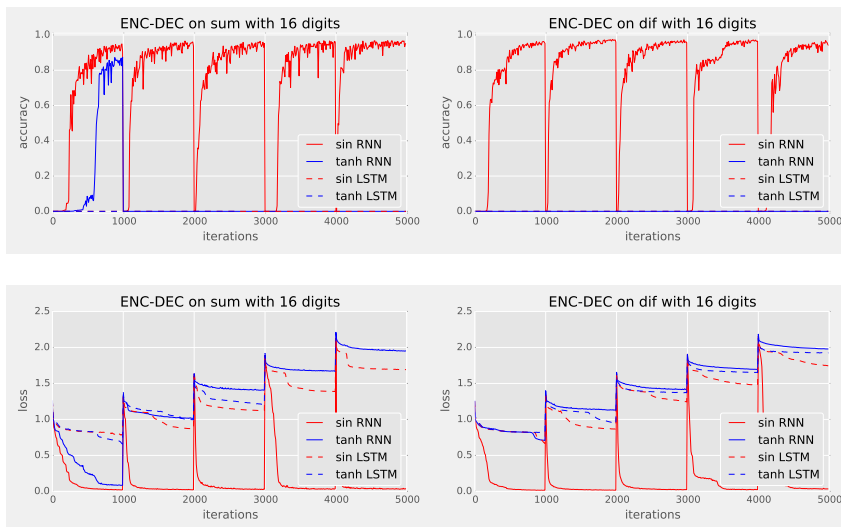

Figure 9: Accuracy and loss curves of the ENC-DEC RNN using sine or tanh, for the tasks *sum* and *dif*. $D$ starts from 8 and increases by 2 every 1000 iterations until it reaches 16 digits per addend at iteration 4000.

