# Peer review of "Taming the waves: sine as activation function in deep neural networks"

_ICLR 2017 — rejected_

[Official Review · AnonReviewer4 · rating 4 · confidence 4 · 16 Dec 2016]

An interesting study of using Sine as activation function showing successful training of models using Sine. However the scope of tasks this is applied to is a bit too limited to be convincing. Maybe showing good results on more important tasks in addition to current toy tasks would make a stronger case?

[Official Review · AnonReviewer3 · rating 4 · confidence 4 · 19 Dec 2016]
**Mainly theoretical idea with insufficient evidence of being practical**

Authors propose using periodic activation functions (sin) instead of tanh for gradient descent training of neural networks.
This change goes against common sense and there would need to be strong evidence to show that it's a good idea in practice. 
The experiments show slight improvement (98.0 -> 98.1) for some MNIST configurations. They show strong improvement (almost 100% higher accuracy after 1500 iterations) on a toy algorithmic task. It's not clear that this activation function is good for a broad class of algorithmic tasks or just for the two they present. Hence evidence shown is insufficient to be convincing that this is a good idea for practical tasks.

[Official Review · AnonReviewer2 · rating 4 · confidence 4 · 20 Dec 2016]
**Nice preliminary theoretical results of using sin activations, but more evidence needed**

Summary:
In this paper, the authors explore the advantages/disadvantages of using a sin activation function.
They first demonstrate that even with simple tasks, using sin activations can result in complex to optimize loss functions.
They then compare networks trained with different activations on the MNIST dataset, and discover that the periodicity of the sin activation is not necessary for learning the task well.
They then try different algorithmic tasks, where the periodicity of the functions is helpful.

Pros:
The closed form derivations of the loss surface were interesting to see, and the clarity of tone on the advantages *and* disadvantages was educational.

Cons: 
Seems like more of a preliminary investigation of the potential benefits of sin, and more evidence (to support or in contrary) is needed to conclude anything significant -- the results on MNIST seem to indicate truncated sin is just as good, and while it is interesting that tanh maybe uses more of the saturated part, the two seem relatively interchangeable. The toy algorithmic tasks are hard to conclude something concrete from.

[Author Response · Giambattista Parascandolo · 13 Jan 2017]
**Comment to all reviewers**

Thank you to all the reviewers for the helpful comments. Probably due to a lack of clarity of the paper, some details in the proposed summaries differ from the main points we tried to convey. Let us then attempt to clarify what the point of the paper is supposed to be, both here and in the manuscript.

The main point of the paper is to analyze the effect of sine as an activation function, how it affects the representation learned and what consequences it has on learning. 

The main contributions of the paper are:
O1 - a proof that confirms the claim that networks with sine might easily get stuck into local minima, obtained by analytically characterizing the loss surface.
O2 - evidence showing that when training is successful for typical datasets (as it's the case for most previous works using sine presented in Section 2) the network is actually not relying on the periodicity of the function. The network relies almost exclusively on the central part of the function, that is monotonic and similar to a tanh. Evidence for this is presented on MNIST and in the new version on Reuters too.
O3 - there might still be room for sine as activation function in certain artificial/algorithmic tasks where sine is intuitively beneficial, since there are at least two simple tasks (i.e. sum and diff) where sin can outperform a standard RNN baseline using tanh.

We are trying to make these points more clear and explicit in the paper.

[Final Decision · Program Chairs · 06 Feb 2017]
**ICLR committee final decision**

The reviewers unanimously recommend rejecting the paper.